# A Participation Degree-Based Fault Detection Method for Wireless Sensor Networks

**DOI:** 10.3390/s19071522

**Published:** 2019-03-28

**Authors:** Wei Zhang, Gongxuan Zhang, Xiaohui Chen, Xiumin Zhou, Yueqi Liu, Junlong Zhou

**Affiliations:** 1Computer Science and Engineering, Nanjing University of Science and Technology, NO. 200 Xiaolingwei Road, Nanjing 210094, China; zw@hytc.edu.cn (W.Z.); cxhlcx@hytc.edu.cn (X.C.); 312062290@njust.edu.cn (X.Z.); lyq@hytc.edu.cn (Y.L.); jlzhou@njust.edu.cn (J.Z.); 2Computer Science and Technology, Huaiyin Normal University, NO. 111 Changjiangxi Road, Huai’an 223300, China

**Keywords:** outlier detection, fault detection, participation degree, hierarchical clustering, WSNs

## Abstract

In wireless sensor networks (WSNs), there are many challenges for outlier detection, such as fault detection, fraud detection, intrusion detection, and so on. In this paper, the participation degree of instances in the hierarchical clustering process infers the relationship between instances. However, most of the existing algorithms ignore such information. Thus, we propose a novel fault detection technique based on the participation degree, called fault detection based on participation degree (FDP). Our algorithm has the following advantages. First, it does not need data training in labeled datasets; in fact, it uses the participation degree to measure the differences between fault points and normal points without setting distance or density parameters. Second, FDP can detect global outliers without local cluster influence. Experimental results demonstrate the performance of our approach by applying it to synthetic and real-world datasets and contrasting it with four well-known techniques: isolation forest (IF), local outlier factor (LOF), one-class support vector machine (OCS), and robust covariance (RC).

## 1. Introduction

Due to advances in the electronic industry, sensors are employed in different aspects of our daily life [1]. By increasing the usage of sensors [2], the measurements from sensors become critical in decision-making tasks and important in the accuracy of data measured by the sensors [3]. In WSNs, faults are defined as “metrics that deviate significantly from normal sensed data patterns” [4]. Outliers are anomalous objects that lie far away from other data points on a scatter plot of the data. The detection of outliers has significant relevance and often provides critical actionable information in many domains. The characteristics of interest or real-life relevance of outliers is key to outlier detection. Outlier detection techniques provide associated definitions of an anomaly and there exists some overlap between them [5].

Model-based techniques build a model of the data that outliers do not fit very well. Schölkopf et al. (2001) proposed the one-class support vector machine (SVM) [6] which is an extension of the support vector algorithm for outlier detection. Local correlation integral (LOCI) [7] is another model-based outlier detection method based on probabilistic reasoning which has been explored by Papadimitriou et al. (2003). Nguyen et al. (2010) used robust covariance estimation [8] to pull the maximum likelihood estimation (MLE) toward outliers by dominating the log likelihood function. Isolation forest, or iForest [9], is a different type of model-based method that isolates anomalies from profiles of normal instances described by Niu et al. (2012). The cluster merging system (CMS) [10] proposed by de Morsier et al. (2015) has a new cluster validity measure (CVM) that quantifies the clustering performance of hierarchical algorithms in the presence of outliers. Mokhtar et al. (2018) applied the Caires and Wyatt model to the hierarchical clustering procedure for outlier detection [11]. In the fluid distribution systems, automatic fault monitoring and diagnosis is of great relevance worldwide. Recently, Santos–Ruiz et al. (2018) implemented a dynamic principal component analysis (DPCA) [12] model to detect fluid leaks in an experimental pipeline by training the measurement data.

Distance-based or proximity-based techniques define a proximity measure between objects. Anomalous objects are those that are distant from most of the other objects. Knorr et al. (1998) studied on distance-based outliers [13] and provided formal and empirical evidence justifications for finding distance-based outliers. Then Knorr et al. (2000) improved the study on distance-based outliers to make the algorithm efficient for large, k-dimensional datasets [14]. In the same year, Ramaswamy et al. (2000) proposed a novel formulation for distance-based outliers [15] to find outliers by labeling the top points in distance ranking as outliers. In another way, Niu et al. (2007) described outlier detection using distribution clustering (ODDC) [16] which clusters in the distribution difference space rather than in the original feature space.

Density-based techniques estimate the density of objects. Objects that are in regions of low density are relatively distant from their neighbors and can be considered anomalous. Breunig et al. (2000) represented a local outlier factor (LOF) [17] which is a density-based local outlier detection method in which an outlier identification degree is assigned to objects. A biased sampling method [18] was used by Kollios et al. (2003) as a general technique for density-biased sampling and employs clustering and outlier detection algorithms. Based on the extended nearest neighbors of the object, Tang et al. (2017) introduced a relative density-based outlier score (RDOS) [19] to measure the local outlierness of objects which estimates the density distribution with a local kernel density estimation (KDE) method.

Clustering-based techniques find groups of strongly related objects, while outlier detection finds objects that are not strongly related to other objects. Almeida et al. (2007) modified the hierarchical cluster analysis (HCA) outlier detection method [20] and improved the single linkage hierarchical cluster analysis so as to circumvent the sensitivity to outliers. Jiang et al., (2008) studied on clustering-based outlier detection (CBOD) [21] which clusters the dataset with a one-pass clustering algorithm and determines the outlier cluster by the outlier factor. Krishnamoorthy et al. 2016 improved alimited iteration agglomerative clustering (iLIAC) [22] which works with a new threshold (optimum merge cost) to limit the number of iterations and automatically identifies the highly relative clusters and outliers on a large dataset. Gullo et al. (2017) described a prototype-based agglomerative hierarchical clustering method called U-AHC [23], which uses new uncertain linkage criteria for clustering.

In WSNs, faults are anomalies that differ from other points in many features [24]. However, due to the limited capabilities of sensor nodes, sensor observations collected from sensor nodes typically have lower data quality and reliability. There are many difficulties and challenges of fault detection in WSNs. In this paper, we propose a novel fault detection method based on the participation degree, called fault detection based on participation degree (FDP), to overcome the shortage of some well known outlier detection algorithms in fault detection for WSNs. The major contributions of this paper are summarized as follows.
We propose the FDP algorithm, which uses a new concept, the participation degree, as a measure for detecting anomalous objects. The FDP algorithm is very different from the existing outlier detection which needs to select and optimize the distribution model, distance parameters, or density parameters in the process of outlier detection.In the FDP algorithm, we process data based on the nearest neighbor boundary (NNB) [25], which greatly reduces the time complexity and space complexity of our algorithm. This allows the FDP algorithm to process actual big data. Additionally, based on NNB, the data can be stored distributively, and processed, so that the FDP algorithm has the ability to process massive data.In hierarchical clustering, we use different similarity measures, such as single link, complete link, group average, and Ward’s method [26,27,28],and the FDP algorithm obtains more accurate outlier detection results.We analyze the performance of the FDP algorithm and compare to traditional methods using synthetic data and real data in some experiments. The experimental results show that the performance of the FDP algorithm has better outlier detection than the other compared algorithms.

The rest of the paper is structured as follows. Section 2 gives the related works for our fault detection method. Section 3 presents the concept of the FDP algorithm and the theoretical proof of fault detection metrics based on the participation degree. In Section 4, we describe the evaluation results of our proposed approach. Finally, Section 5 concludes the paper.

## 2. Background

Based on the hierarchical clustering method, the FDP algorithm adopts the similarity measurement method [26,27,28,29] in clustering, uses the quad-tree [30,31,32] and kd-tree [33,34] structures to store data, and employs NNB [25] to slice data. Cluster aggregate inequality [27] guarantees the correctness of the results. This section provides a brief introduction to the relevant previous research.

### 2.1. Similarity in Hierarchical Clustering

A similarity measure between objects is the basis of all clustering algorithms. A common similar distance measure between individual objects is the Euclidean distance, but when there are multiple objects in clusters, some similarity calculations of the distance between the clusters are required. The similarity function is an important measure of similarity between two hierarchical clusters.

The MIN (also called single link), MAX (also called complete link), group average, Ward’s method, and centroid method are common similarity functions [26,27,28]. Single link uses the minimum distance of objects in clusters *A* and *B* to represent the proximity between clusters *A* and *B*. The similarity function of MIN is formulated as
(1)$$Sim_{min}\left(A,B\right)=\min_{a\in A,b\in B}dist\left(a,b\right),$$
where dist(a,b) denotes the Euclidean distance between objects *a* and *b*. The complete link method uses the maximum distance between objects in clusters *A* and *B*. The similarity function of MAX is
(2)$$Sim_{max}\left(A,B\right)=\max_{a\in A,a\in B}dist\left(a,b\right).$$

The group average is the average distance between pairs of objects in clusters *A* and *B*. The similarity function of the group average is described as
(3)$$Sim_{avg}\left(A,B\right)=\sum_{a\in A,b\in B}\frac{dist(a,b)}{n_{A}\times n_{B}},$$
where $n_{A}$ and $n_{B}$ denote the numbers of objects in clusters *A* and *B*, respectively. The centroid method uses the distance between the centroids of clusters *A* and *B*. The similarity function of the centroid method is
(4)$$Sim_{centroid}\left(A,B\right)=dist\left(c_{A},c_{B}\right),$$
where $c_{A}$ and $c_{B}$ denote the centroids of clusters *A* and *B*, respectively. The centroid of a cluster is the arithmetic mean position of all the points in the cluster. The similarity function of Ward’s method is
(5)$$Sim_{ward}\left(A,B\right)=\frac{dist(c_{A},c_{B})}{\frac{1}{n_{A}}+\frac{1}{n_{B}}},$$
where $n_{A}$ and $n_{B}$ denote the numbers of objects in clusters *A* and *B*, respectively. $c_{A}$ and $c_{B}$ denote the centroids of clusters *A* and *B*.

### 2.2. Quad-Tree and kd-Tree

The quad-tree [30,31,32] and kd-tree [33,34] are important data structures. Quad-tree organizes points in a two-dimensional space that can divide the two-dimensional data into subspaces, thus one can store and manage the two-dimensional data.The kd-tree is a binary tree that can store and manage multidimensional data. Each non-leaf node in the kd-tree is divided into two parts by using a split hyperplane. By calculating the range of the NNB [25] in the storage partition of the quad-tree or kd-tree and in the nearest neighbor search of the NNB can effectively reduce the algorithm complexity.

### 2.3. Nearest Neighbor Boundary

The concept of the NNB [25] is a region in which a point can find all of its nearest neighbors. NNB can greatly reduce the scope of finding the nearest neighbors and greatly improves the efficiency of the FDP algorithm. The definition is given below.

**Definition** **1.**
*Given a rectangle region $R_{1}$, $A(R_{left},R_{bottom})$ is the coordinates of the lower left corner of the region $R_{1}$, $B(R_{right},R_{top})$ is the coordinates of the upper right corner of the region $R_{1}$. The NNB of region $R_{1}$ is a rectangle defined as follows. $NNB\left(R_{1}\right)=\left\{\left(x,y\right)\right|BR_{left}-dist\left(A,B\right)\leq x\leq R_{right}+dist\left(A,B\right)$, $R_{bottom}-dist\left(A,B\right)\leq y\leq R_{top}+dist\left(A,B\right)\}$.*


From the definition of NNB, the nearest neighbors of all of the points in region $R_{1}$ are included in $NNB(R_{1})$ [25]. By using NNB, a large dataset can be divided into independent subsets and then the nearest neighbor of each point in each subset is found. Based on NNB, the time complexity of the FDP algorithm can be reduced to $O(n\log^{2}n)$, where *n* is the datasize of the dataset.

### 2.4. Mutual Nearest Neighbor and Cluster Aggregate Inequality

In our algorithm, the mutual nearest neighbor and cluster aggregate inequality are important concepts used to ensure the correctness of the algorithm. Mutual nearest neighbors (MNs) [35] are pairs of object *A* and *B*, such that *B* is the nearest neighbor of *A* and *A* is the nearest neighbor of *B*. If a similarity measure conforms to the cluster aggregate inequality [35] it is guaranteed that clustering can be directly completed with nearest neighbors, and the final clustering result is consistent with the clustering result of the original agglomerative hierarchical clustering (AHC) algorithm. The cluster aggregate inequality is described as follows.
(6)Sim(A∪B,C)≥min(Sim(A,C),Sim(B,C)).

The single link, complete link, group average, and Ward’s similarity measures all satisfy the cluster aggregate inequality [27]. After the FDP algorithm has managed data using the quad-tree or kd-tree, it searches for all mutual-nearest neighbors by single link, complete link, group average, and Ward’s method in the NNB and merges them in each iteration. The cluster aggregate inequality guarantees that the FDP algorithm is correct even when the algorithm complexity is reduced.

## 3. Fault Detection Based on the Participation Degree

As mentioned above, an outlier is an object that is significantly different from most other objects. A good outlier detector must have an effective metric to identify the differences between objects. This section describes the detection of outliers based on the degree of participation of objects.

### 3.1. Motivation Example

Agglomerative hierarchical clustering (AHC) is a classic clustering analysis method that clusters items into a whole component and constructs a cluster hierarchy that reflects the relationships between items. The clustering method uses some similarity measures to combine similar items to form clusters that items in the same cluster are very similar. The AHC algorithm is mainly applied to classification which is used in some traditional cluster-based outlier detection algorithms (such as CBOD [21], U-AHC [23], iLIAC [22], etc.) to detect anomalies. However, in our research, it was found that by observing the process of hierarchical clustering, rather than the classification result, the measurement of new abnormal points can be extracted, and abnormality detection can be more effectively performed. A simple example of the new method for explaining an detected abnormal point is shown in Figure 1.

In Figure 1a, there are twelve points, A to L. We know that the points that are close to each other are more likely in a cluster. It is observed that six points, A–F, formed a group relatively close to each other. The two individuals J and K form a small group, and G, H, I, and L were far away from others. In fact, these six points, J to L, were abnormal points. In the hierarchical clustering analysis, merging was performed according to the similar distance between the data. Hierarchical clustering analysis found the two closest points, B and C, and merges them into a new cluster, M. In the following clustering process, cluster M represents B and C for the distance calculation and then merges with other points or clusters. This process is repeated until all points or clusters are merged into one cluster W. We record this merging process to form a dendrogram, as shown in Figure 1b. In the figure, we can see that the hierarchical clustering finds the nearest pair and merges them to naturally form a classification.

In Figure 1a, the participation degree of each point is marked in the upper right corner of each point. It can be seen from the figure that points A–F are insiders, and the others are far from insiders; they are outliers. From Figure 1, it can be observed that in the hierarchical clustering process, each point participates in a number of merging processes and this has a correlation with whether or not the point is an abnormal point. According to the statistical probability of the abnormal points of the dataset and the statistics of the number of steps each point participates in the hierarchical clustering process, an abnormal point can be determined. For a 50% probability that an abnormal point is in the dataset, a threshold value of 4 is taken as the abnormal point judgement standard, and points G, H, I, J, K, and L are determined as abnormal points. Moreover, it can be found that points J and K are the core points of cluster U, but their participation degrees are much lower than the threshold; points G and H are not core points in class T although they are merged more frequently.

As described above, based on the participation degree, we can determine the steps when each object participates in the clustering merging process. A metric of participation can be used as an important measure in the detection of abnormal points. We define a new, efficient metric to detect abnormal points, the participation degree, which describes the steps that each object participates in the clustering merging process. The traditional method can also consider a cluster of a small number of people to be an abnormal point according to this classification. However, in our research, it was found that in this clustering process, the anomaly points have the characteristic of a smaller participation degree. As shown in Figure 1a, in the hierarchical clustering process, outliers have less participation degree than insiders. This feature can identify the abnormal points that cannot be accurately recognized by the traditional clustering algorithm for outlier detection.

### 3.2. Participation Degree

From the above Section 3.1, a binary tree structure called a dendrogram tree is generated during the merging process of hierarchical clustering. Additionally, through Figure 1b, if an object is merged more frequently than other objects, the object is mostly a normal point. If the merging participation of an object is small, then the object is usually an exception. In this subsection, we first give some definitions for the merge distance and the participation degree. Based on these definitions, we give a theorem and prove that the participation degree can be used as an abnormality metric.

First, some items are given. If there are *n* objects for hierarchical clustering, the resulting binary tree has 2n−1 nodes, also called 2n−1 clusters. The *n* nodes are the original clusters (A–L), which are the leaf nodes in the binary tree. The following n−1 nodes, merged from other clusters, are new clusters, which are intermediate nodes (M–W) in the binary tree, as shown in Figure 1b.

The traditional hierarchical clustering algorithm finds the pair of objects with the closest similarity distance, merges them, and thus replaces them with a new cluster. Then, it calculates the similarity distance between the new object and other objects or clusters and finds the closest pair. The closest clusters are merged until at the end, one cluster is left. The similarity distance recorded when the object or cluster is merged is the merge distance, which has the following definition.

**Definition** **2.**
*Given a cluster set $X=\{x_{1},x_{2},\ldots ,x_{2n-1}\}$ in a hierarchical clustering, let $x_{i}$ and $x_{j}$ be nearest clusters that are merged to form a new cluster. $MergeDistance(x_{i})$ and $MergeDistance(x_{j})$ are defined as the similarity distance between $x_{i}$ and $x_{j}$ when they are merged.*
$$MergeDistance(x_{i})=MergeDistance(x_{i})=Sim(x_{i},x_{j}),$$
*where Sim() is a similarity function, as shown in Section 2.1, such as MIN (single link), MAX (complete link), group average, or Ward’s method. It should be noted that the similarity functions used in the FDP algorithm must satisfy the cluster aggregate inequality [27].*


Single link, complete link, group average, and Ward’s method satisfy the cluster aggregate inequality, as described in Section 2.4. The cluster aggregate inequality guarantees that the FDP algorithm, which adopts the participation degree as the abnormality metric, can correctly detect outliers. This is discussed after the proof of Theorem 1.

By observing the example in Section 3.1, it can be seen that the participation degree of each object is the same as the level of each cluster in the dendrogram tree. This is because in the hierarchical clustering process, when two clusters are merged, the participation degrees of the original clusters that belong to the two merged clusters are incremented; the numbers of levels increase in the binary tree. Therefore, for measurement of fault detection, the participation degree, is defined as follows.

**Definition** **3.**
*Give a cluster set $X=\{x_{1},x_{2},\ldots ,x_{2n-1}\}$, let T be the binary tree generated from the dendrogram of the hierarchical clustering process. The participation degree of a cluster $x_{i}$ in the binary tree T is*
$$ParDegree\left(x_{i}\right)=level\left(x_{i},T\right),$$
*where $level(x_{i},T)$, i=1,2,…,2n−1 is the level of node $x_{i}$ in the tree T.*


From Definition 3, the participation degree of a cluster can be calculated by computing the cluster level in the dendrogram tree from hierarchical clustering. The participation degree of objects is used to measure the abnormality by the FDP algorithm. Below, we prove the theory that the participation degree can be used as a metric for outlier detection. According to the definitions of the participation degree and merge distance, we have the following theorem.

**Theorem** **1.**
*Given a cluster set $X=\{x_{1},x_{2},\ldots ,x_{2n-1}\}$ in a hierarchical clustering, for $x_{i},x_{j}\in X$, if $ParDegree\left(x_{i}\right)>ParDegree\left(x_{j}\right)$, the $\exists x_{k}\in X$, such that $x_{k}$ is closer to $x_{i}$ than $x_{j}$.*


Theorem 1 shows that in a hierarchical clustering dendrogram tree, if two objects have different participation degrees, the high participation degree object is closer to other objects than the lower participation degree object. In other words, objects with a lower participation degree are relatively far away from objects with a higher participation degree (insiders). The proof of Theorem 1 is as follows.

**Proof** **of** **Theorem** **1**Given a cluster set $X=\{x_{1},x_{2},\ldots ,x_{2n-1}\}$, according to the process of the hierarchical clustering algorithm, if $ParDegree(x_{i})=l$, then $\exists x_{k}$ such that $ParDegree(x_{k})=l$ and $MergeDistance\left(x_{i}\right)=MergeDistance\left(x_{k}\right)$; $x_{k}$ is the nearest neighbor of $x_{i}$. Thus, it is proved. □

From Definition 2, the $MergeDistance(x_{i})$ is the similarity of $x_{i}$ when $x_{i}$ is merged with its nearest neighbor $x_{k}$. The points $x_{i}$ and $x_{k}$ have the same MergeDistance(). It should be noted that MergeDistance() must satisfy the cluster aggregate inequality [27]. The concept of the cluster aggregate inequality is that if there are three clusters, *A*, *B* and *C*, and the nearest neighbors *A* and *B* are merged to a new cluster *D*. Then, the MergeDistance(D,C) is larger than the minimum of MergeDistance(A,C) and MergeDistance(B,C). In other words, the similarity distance of the new cluster produced by the merging of other clusters will monotonically increase. The cluster aggregate inequality guarantees that the FDP algorithm, which adopts the participation degree as the abnormality metric, can correctly detect outliers. Conversely, if a similarity measure does not satisfy the cluster aggregate inequality, then the similarity distance may be reduced between the merged new cluster and other clusters. With such a metric, the proof of Theorem 1 does not hold.

From the proof of Theorem 1, objects with a low participation degree are more likely to be abnormal points. Using this kind of abnormal point metric, the relativity of fault detection is not only partial, but also global. As shown in Figure 1, some anomalies that have been merged into small clusters can also be detected because the overall participation degree of them is low. Hence, the FDP method can detect outliers that cannot be detected by conventional outlier detection algorithms.

### 3.3. The FDP Algorithm

Using the participation degree, we constructed the FDP algorithm based on the agglomerative hierarchical clustering algorithm. As shown in Figure 2, the FDP algorithm is divided into four parts: (i) create the dendrogram tree, (ii) calculate the participation degree, (iii) calculate the threshold degree, and (iv) detect the outliers. The detail of FDP algorithm is shown in Algorithm 1.

In Algorithm 1, a dendrogram tree is created during the clustering process with the similarity functions described in Section 2.1, as shown in Line 4. Then, the participation degree of each node is calculated, and a dictionary of participation degree statistics is returned, as shown in Line 5. A threshold participation degree for detecting outliers is computed and returned, as shown in Line 6. In the end, by comparing with the threshold participation degree, outliers in the dataset are labeled, as shown in Line 7. The details of these four steps are described below.
**Algorithm 1:** The fault detection based on participation degree (FDP) algorithm.
**Input**: dataset—outlier detection dataset, outFraction—outlier probability ratio,
   linkage—similarity function**1**initialize treeRoot—the tree root node;**2**initialize degDictionary—dictionary of participation degree statistic;**3**initialize threshold—threshold participation degree;**4**CreDendTree(dataset,treeRoot,linkage) //see Algorithm 2;**5**CalParDegree(treeRoot,−1,degDictionary) //see Algorithm 3;**6**CalThrDegree(degDictionary,outFraction,threshold) //see Algorithm 4;**7**DetectOutlier(treeRoot,threshold) //see Algorithm 5;

In Algorithm 2, the time complexity of building a dendrogram tree based on the basic hierarchical clustering algorithm is $O(n^{3})$. The “create dendrogram tree” procedure can be reduced to $O(n\log^{2}n)$ by dataset partition. First, the data is stored in a quad-tree or kd-tree, as shown in Line 5. Then, the data is grouped by the NNBs of each region, as shown in Line 6. After grouping data, searches for all mutual nearest neighbors are conducted by single link, complete link, group average, or Ward’s method in the NNB. The tasks of computing the nearest neighbor pair in each group are independent from each other, as shown in Line 7. Global mutual nearest neighbor pairs are calculated from proximity matrices, as shown in Line 8. All mutual nearest neighbors are merged in each iteration, as shown in Lines 9 to 12. At the end of Algorithm 2, the top node is returned as the root of the dendrogram tree.
**Algorithm 2:** Create dendrogram tree.
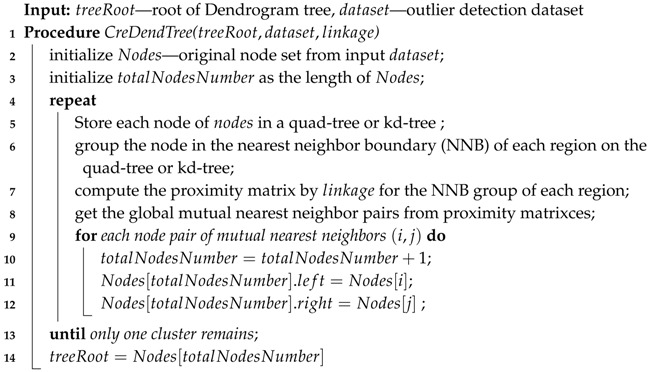


In Algorithm 3, there is a recursive procedure CalParDegree(). When we invoke CalParDegree() in Algorithm 1, Algorithm 3 calculates all nodes’ participation degrees by the recursive calling procedure CalParDegree() and records the number of the original data points at each participation degrees in degDictionary.
**Algorithm 3:** Calculate participation degree.
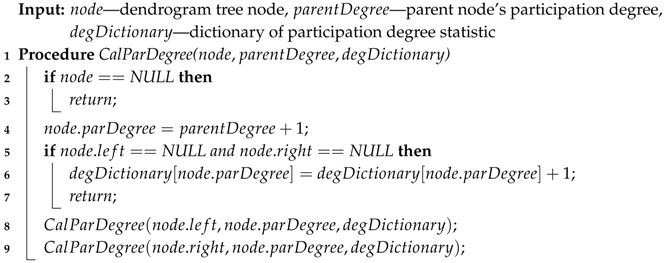


In Algorithm 4, the threshold of the participation degree is calculated from the degDictionary and outFraction. First, totalNumber, the total amount of data, is computed by adding up all the items’ values in degDictionary, as shown in Lines 4 and 5. Then, outlierSum, the outlier amount, is computed by adding up the items’ values in degDictionary from low key to high key. When the outlierSum is more than the outFraction of the totalNumber, the threshold of the participation degree is obtained as the item key of the degDictionary, as shown in Lines 6 to 10.
**Algorithm 4:** Calculate threshold degree.
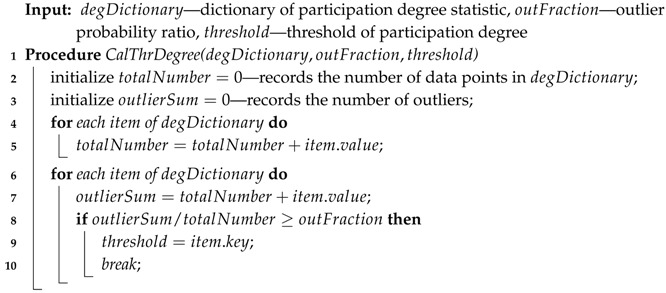


In Algorithm 5, there is a recursive procedure, DetectOutlier(). When we invoke DetectOutlier() in Algorithm 1, Algorithm 5 labels nodes whose participation degree is below or equal to the threshold by the recursive calling procedure DetectOutlier(), as shown in Lines 4 and 5. Finally, the FDP algorithm completes outlier detection.
**Algorithm 5:** Detect outlier.
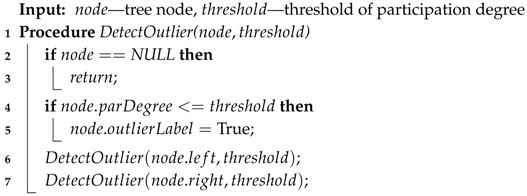


### 3.4. Algorithm Complexity Analysis

The functions CreDendTree(), CalParDegree(), CalThrDegree(), and DetectOutlier() are four important procedures of the FDP algorithm, as shown in Algorithm 1. The procedure CreDendTree() uses the quad-tree or kd-tree to store the dataset, and it uses a binary tree to store the dendrogram tree. The storage space complexity of the quad-tree or kd-tree and the binary tree is O(n+2n−1) = O(n), where *n* is the number of data items in the dataset. The space complexity for computing the proximity matrix for the NNB group of each region is $O(m^{2})$, where *m* is the maximum number of points in the leaf node, as shown in Algorithm 2. In practice, *m* is much smaller than *n*, so the total space complexity of the CreDendTree() procedure is O(n). The CalParDegree(), CalThrDegree(), and DetectOutlier() procedures have a storage space complexity of O(n), as shown in Algorithms 3–5, respectively. Adding these up, the storage space complexity of the FDP algorithm is O(n) as shown in Table 1.

For Algorithm 2, the procedure CreDendTree(), the time complexity of inserting points in a quad-tree or a kd-tree is $O\left(n\log\frac{n}{m}\right)$. In a quad-tree or kd-tree, partitioned regions of leaf nodes do not overlap each other. Therefore, the time complexity of grouping data is $O\left(\frac{n}{m}\log\frac{n}{m}\right)$, as shown in Line 6. When computing the proximity matrix for mutual nearest neighbor pairs in the NNB group of each region, the time complexity is $O(m^{2})$, as shown in Line 7. The time complexity of computing the proximity matrix in all regions is $O\left(\frac{n}{m}m^{2}\right)=O\left(nm\right)$. The time complexity of each iteration in the procedure CreDendTree() is $O(nm+\left(n+\frac{n}{m}\right)\log\frac{n}{m})$. In practice, *m* is much smaller than *n* so the time complexity is O(nlogn) in an iteration. Additionally, the time for iteration is about logn, so the total time complexity of procedure CreDendTree() is $O(n\log^{2}n)$. The time complexity of the other three procedures, CalParDegree(), CalThrDegree(), and DetectOutlier() are same as O(n). So, the time complexity of the FDP algorithm is $O(n\log^{2}n)$ as show in Table 1.

## 4. Evaluation

In this section, the FDP algorithm is performed on synthetic datasets and real-world datasets. In the synthetic dataset experiments, we applied outlier detection algorithms to a series of datasets of different data sizes in order to evaluate the performance of the algorithm as the data size increased. In the real-world dataset experiments, we applied outlier detection algorithms to real-world datasets in order to compare the outlier detection performance between different algorithms.

### 4.1. Compared Outlier Detection Algorithms

In the research of traditional outlier detection algorithms, most research has focused on improving the accuracy of the algorithm. Some commonly used anomaly algorithms are described and analyzed below and compared to the performance of the FDP algorithm in the experiments.

Isolation forest, or iForest [9], is a different type of model-based method that isolates anomalies from profiles of normal instances. The concept of isolation enables iForest to exploit sub-sampling that is not feasible with other methods. The iForest algorithm has a linear time complexity and a low space complexity. iForest works well in datasets with high dimensionality and in situations of training in a dataset without any anomalies. The overall average time complexity of the iForest algorithm is $O(t\psi^{2}+nt\psi )$, where ψ is subsampling size, *t* is the number of trees and *n* is the testing data size.

LOF [17] uses the local anomaly factor to indicate the outlierness degree of an object. LOF is local and depends on the degree of isolation of an object from its surrounding neighborhood. LOF is defined as the ratio of the local reachability density and the average local reachability density of its k-nearest neighbors. LOF has many desirable properties and can be used to find meaningful outliers, but not by existing methods. The time complexity of the LOF algorithm is $O(n^{2})$.

The one-class SVM [6] classifies objects into outliers and normalities by finding a hyperplane between them. The one-class SVM algorithm is an extension of the support vector algorithm for the case of outlier detection. The time complexity of the one-class SVM algorithm is $O(n^{3})$ [36].

The robust covariance estimation problem [8] and the outlier detection problem are interchangeable. By using the maximum likelihood estimation (MLE), the robust covariance estimation method estimates data parameters for a known distribution. For outlier detection, the MLE estimators are pulled toward outliers by dominating the log likelihood function. The time complexity of the robust covariance algorithm is $O(n^{3})$.

### 4.2. Experimental Preparation

The experiments were conducted on a server which has a 2.5 GHz Intel Xeon CPU E5-2640 with 24 cores. Our experimental study was focused on the performance of the FDP algorithm compared with isolation forest (IF), local outlier factor (LOF), one-class SVM (OCS), and robust covariance (RC). We adopted the average, complete, single, and Ward’s similarity functions to the FDP algorithm and obtained the FDP average (FDPA), FDP complete (FDPC), FDP single (FDPS), and FDP Ward’s (FDPW) algorithms. The algorithms were performed on synthetic datasets and real-world datasets.

The purpose of synthetic data experiments is to compare the performance of outlier detection algorithms on particular datasets. The synthetic data in experiments is two-dimensional for easy visual display and commonly used in outlier detection. One of the synthetic datasets has two clusters with different distribution radii, and the other has two semi-moon classes, and are summarized in Table 2.

By using the function make_blobs() and make_moons() of Scikit-learn [37], a software machine learning library of Python, we generated a series of synthetic datasets of isotropic Gaussian blobs and two interleaving half circles for the experiments. In order to simulate the fault data generated in WSNs, anomalous points were randomly distributed across the region, as shown in Table 2 and Figure 3.

In Figure 3, Dataset D1 consists of two isotropic Gaussian blob clusters which have different distribution radii. The anomaly points are randomly distributed over the area, and some abnormal points are distributed among the two clusters. Dataset D2 consists of two interleaving half circle clusters. The anomalous points are randomly distributed among the region. Special attention should be paid to a small number of anomalies that are surrounded by two clusters.

On the synthetic datasets, the main evaluation indicators were the detection rate, receiver operating characteristic (ROC) curve [38], and time consumption of the algorithms. According to the result of the abnormality detection algorithm, the points were divided into outlier correct recognition (true positives, TP), outlier error recognition (false negatives, FN), insiders error recognition (false positives, FP), and insiders correct recognition (true negatives, TN). We obtained the following equations.
(7)Totalnumberofpoints=|TP|+|FN|+|FP|+|TN|,
(8)Numberoftrueabnormalpoints=|TP|+|FN|,
(9)Numberoftruenormalpoints=|FP|+|TN|.

According to the above formulas, the accuracy of the abnormal point detection and the accuracy of the normal point detection can be defined as follows.
(10)$$Accuracy\;of\;abnormal\;points\;detection=\frac{\left|TP\right|}{Number\;of\;true\;abnormal\;points}$$
(11)$$Accuracy\;of\;normal\;points\;detection=\frac{\left|FP\right|}{Number\;of\;true\;normal\;points}.$$

The outlier detection rate reflects the performance of the outlier detection algorithm with respect to accurate detection. However, the abnormal point detection algorithm can also obtain the detection rate for the normal point while detecting the abnormal points. Therefore, only looking at the detection rate of the abnormal point cannot fully reflect the performance of the algorithm. Thus, we also used the receiver operating characteristic curve ROC [39] to evaluate the performance of the outlier detection algorithm in the experiments.

In order to test the stability of the algorithms, analogous synthetic datasets to those listed in Table 2 were generated with the same style but with different dataset sizes. Examples of synthetic datasets with 3000 points are shown in Figure 4. The data sizes ranged from 1000 to 10,000 in each synthetic dataset style, as listed in Table 2. On the new synthetic datasets, we evaluated the performance of the algorithms in terms of the outlier detection rate, area under curve (AUC) [38], and runtime. In order to better analyze the stability and overall performance of the algorithm on different sized datasets, box plots [40] were used to display and analyze the outlier detection rate and AUC of the algrithms.

In the UCI machine learning library [41], the wireless indoor positioning data set (WILDS) [42] was used in our real-world dataset experiments for fault detection in WSNs. The original WILDS data collects 2000 measurements of the signal strength of seven WiFi signals on smartphones indoor. The decision variable is one of four rooms. Each item of WILDS is a wireless signal strength observed on a smartphone with a value between −98 and −10. The details of the WILDS dataset are described in the UCI Machine Learning Library [41]. For default detection in WSNs, each item in WILDS is treated as a normal data point. In the real-world dataset experiment, each attribute of WILDS is normalized between 0 and 1. In order to simulate the fault data generated in WSNs, anomalous points are generated and randomly distributed among the region. A series of different anomalous point sizes from 100 to 1000 was used in our experiments. Similar to the experiments of the synthetic datasets, we evaluated the accuracy, AUC, and runtime of the algorithm for the real-world dataset.

### 4.3. Experimental Analysis of the Synthetic Datasets

In the synthetic dataset experiments, each dataset had 255 normal points and 45 abnormal points. The FDPC and FDPW algorithms had the best outlier detection rates for two clusters with standard deviations of 1.5 and 0.3, respectively, for dataset D1. The FDPA detection rate was 0.644 and the FDPS detection rate was 0.622, while the detection rates of the FDPC and FDPW algorithms reached 0.711. The specific detection rates are shown in Table 3.

Figure 5 shows the results of outlier detection in dataset D1. From the figure, we can intuitively observe that the outlier detection rates of the FDP algorithms are better than the outlier detection rates of compared algorithms. In particular, the FDP algorithms can identify some abnormal points inside the clusters.

Figure 6 shows the results of outlier detection on dataset D2. From the figure, we can intuitively observe that the outlier detection rates of the FDP algorithms are better than the outlier detection rates of compared algorithms. In particular, the FDP algorithm can correctly identify the anomaly points surrounding the two half circle clusters. Except for the FDPA detection rate of 0.800, the detection rates of the FDPC, FDPS, and FDPW algorithms reached 0.778, 0.756, and 0.689, respectively. The best outlier detection rate of the other algorithms was 0.600. The specific data is shown in Table 3.

For a more comprehensive analysis of the performance of the FDP algorithms, the ROC curve and the AUC of the outlier detection results were analyzed. The ROC curves and the AUC of the outlier detection algorithms for the synthetic datasets are shown in Figure 7.

As shown in Figure 7, in datasets D1 and D2, the ROC curves of the FDP algorithms were near the best possible prediction, a curve with a point in the upper left corner, at coordinate (0, 1) of the ROC space. In dataset D1, the FDP algorithm had a good result, but so did many of the other approaches. On the other hand, in dataset D2, the performance of the FDP algorithm was significantly higher than that of the other algorithms.

We expanded the data size for the synthetic datasets with data sizes between 1000 and 10,000. Each dataset in Table 2 was expanded into 10 datasets of different sizes, called dataset series 1 and dataset series 2. We used a box plot for the outlier detection rate in each dataset series to analyze the performance of the algorithms. The box plots of the outlier detection rate are shown in Figure 8.

In Figure 8a,b, for all of the FDP algorithms, the outlier detection rates are better than the compared algorithms and the distribution of outlier detection rates is relatively concentrated, indicating that the FDP algorithms are more stable. Moreover, the average values of the outlier detection rates for the FDPA, FDPC, and FDPS algorithms are 0.729, 0.729, and 0.847, respectively, in D1 and D2, and the average value of the abnormality detection rate for the other FDP algorithm (FDPW) is also higher than the compared algorithms.

We also used box plots of AUCs in each dataset series to analyze the performance of the algorithms, as shown in Figure 9. In Figure 9a,b, for all of the FDP algorithms, the AUCs are better than those of the compared algorithms. Additionally the distribution of AUCs is relatively concentrated for the FDP algorithms, indicating that the stability of the FDP algorithms is better. The average values of AUCs of the FDPA and FDPS algorithms are 0.911 and 0.929, respectively, in D1 and D2, and the average value of the abnormality detection rate of the other FDP algorithms is also higher than those of the compared algorithms. The average time cost of all outlier detection algorithms on the series of synthetic datasets is listed in Table 4. It is shown that the running time of the FDP algorithms is not much different from that of the compared algorithms.

Based on the synthetic dataset experiments, we investigated the results of the FDP algorithms and the compared algorithms. We compared the outlier detection rate, AUC, and time cost between algorithms under different sized dataset series and analyzed the results and performances of the FDP algorithms. From the experiments, FDPW outperforms the result in D1 to a greater degree than in D2 because Ward’s method is suitable for detecting blob-based data classification. The results show that the FDP algorithms perform better on synthetic datasets than the compared algorithms when choosing the appropriate similarity functions.

### 4.4. Analysis of Real-World Dataset Experiments

In the experiments on the real-world dataset WILDS described in Section 4.2, we evaluated the accuracy, AUC, and runtime of the FDP algorithms compared to traditional algorithms. We studied WILDS data with different numbers of fault points, between 100 and 1000. We used a box plot of the outlier detection rates and AUCs in real-world datasets to analyze the performance of the algorithms, as shown in Figure 10. It can be observed that the FDP algorithms performed well on the real-world datasets.

In Figure 10a, except for FDPW, the outlier detection rates of FDPS, FDPA, and FDPC are better than the compared algorithms and the distribution of outlier detection rates is relatively concentrated, indicating better stability of these algorithms. Moreover, the average values of the outlier detection rates for the FDPA, FDPC, and FDPS algorithms were 0.948, 0.938, and 0.957, respectively. These rates were higher than the compared algorithms, as shown in Table 5.

For a more comprehensive analysis of the performance of the FDP algorithms, the ROC curves and AUCs of the outlier detection results were analyzed. In Figure 10b, the AUCs of FDPS, FDPA, and FDPC are better than the compared algorithms. Additionally, the distribution of AUCs is relatively concentrated, indicating better stability of these algorithms. Moreover, the average values of AUCs of the FDPA, FDPC, and FDPS algorithms are 0.993, 0.989, and 0.997, respectively. These were higher than the compared algorithms, as shown in Table 6.

Figure 11 shows two example ROC curves for the real-world datasets with 300 and 600 outliers. From the figure, the FDP algorithms have good ROC curves and AUC values for the real-world datasets. The AUC values in the real-world dataset experiments are listed in Table 6.

Table 7 shows that the time consumption of the algorithms on different datasets was different. The OCS and IF algorithms had good operational efficiency on most datasets. As a whole, the FDP algorithm showed little difference compared with the running time of the OCS and IF algorithms. Especially on the datasets with more outliers, the FDPS algorithm ran faster than the compared algorithms. Overall, the performance of the FDP algorithms was efficient.

According to the above real-world dataset experiments of the FDP algorithms, the FDP algorithm performed better than IF, LOF, OCS, and RC when using a proper similarity function. As such, the FDP algorithm could be further improved by using different similarity functions.

## 5. Summary and Future Work

Fault detection is an important method to clarify the accuracy of measurement data in WSNs. In practice, it is difficult to determine the accuracy of the measured data by WSNs.In this paper, we implemented a new fault detection method based on the participation degree, called FDP. This measurement visually reflects the relationship between objects that can detect anomalous objects with high accuracy. Furthermore, the FDP algorithm does not require manual settings of parameters such as model, distance or density. FDP is an unsupervised fault detection algorithm that does not require a training set to build the model, so it is very suitable for fault detection in WSNs. In the algorithm, the nearest neighbor boundary (NNB) is used to slice the data, which greatly reduces the time complexity of the FDP algorithm for processing large data sets. We conducted a number of experiments on synthetic and real-world datasets to evaluate the effectiveness of our algorithm. Experimental results show that the performance of the FDP algorithm outperforms other benchmark algorithms.

There are two main directions for our future works. First, the application of similarity measurement functions in the FDP algorithm will be studied to solve various practical problems in particular environments with wireless sensor networks. Second, the FDP algorithm will be implemented in other distributed platforms for processing massive data.

## Figures and Tables

**Figure 1 sensors-19-01522-f001:**
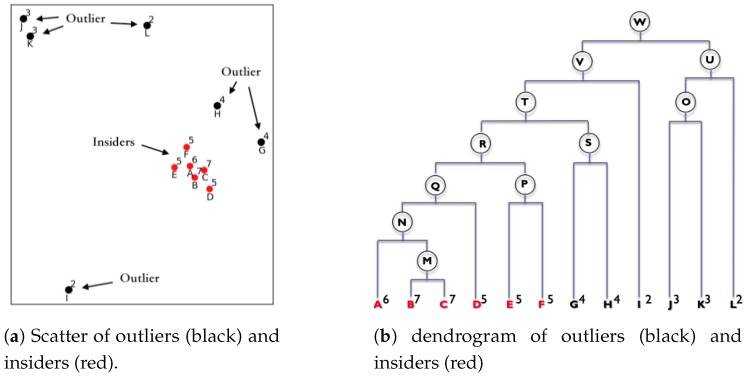
An example of clustering with several outliers: according to the hierarchical clustering result, the participation degrees are marked for each point.

**Figure 2 sensors-19-01522-f002:**
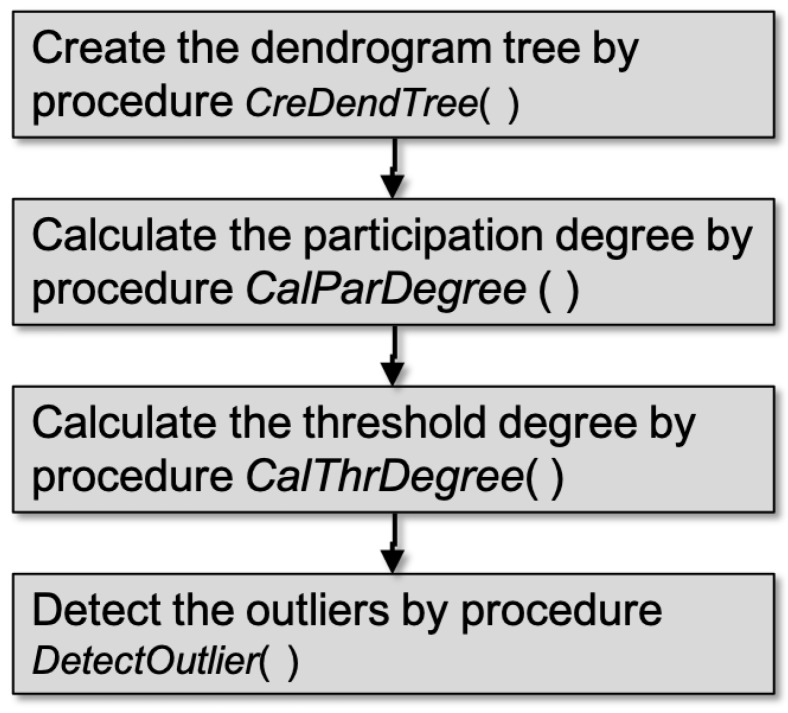
Schematic diagram of the fault detection based on participation degree (FDP) algorithm.

**Figure 3 sensors-19-01522-f003:**
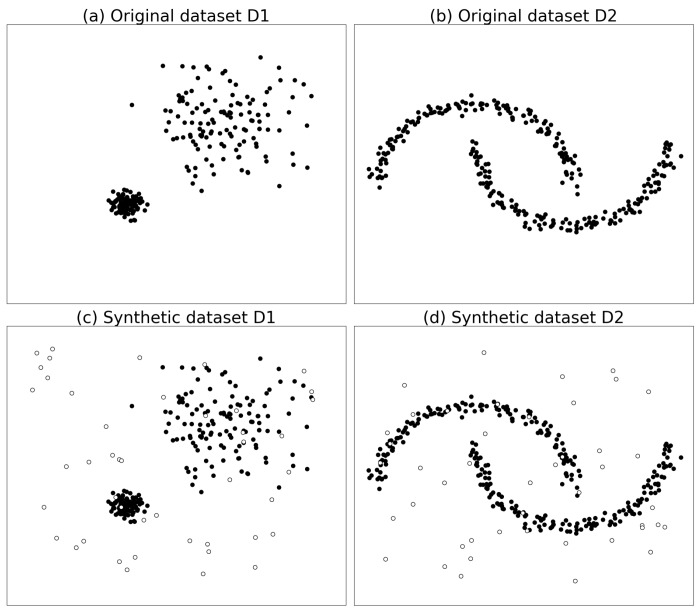
Synthetic datasets in Table 2, the upper frames represent the original clusters (**black**) in the datasets D1 and D2, and the lower frames represent the synthetic datasets with outliers (**white**) generated by a Gaussian distribution.

**Figure 4 sensors-19-01522-f004:**
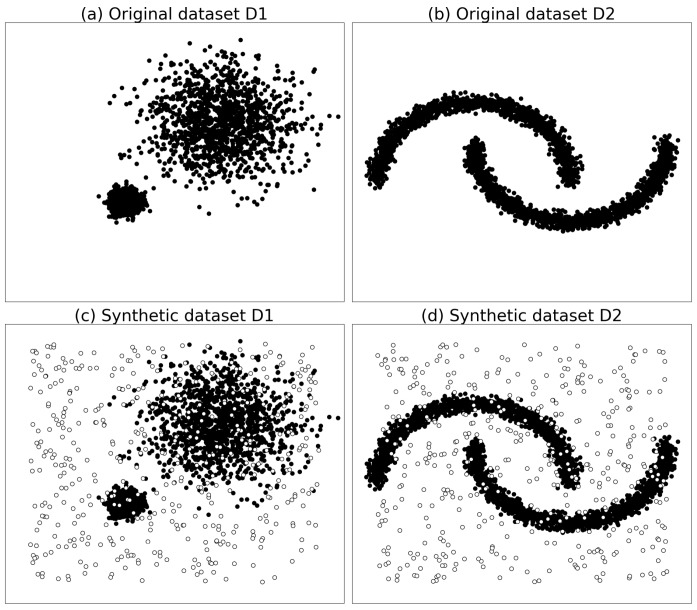
Examples of synthetic datasets analogous to those in Figure 3 but with 3000 points in each dataset.

**Figure 5 sensors-19-01522-f005:**
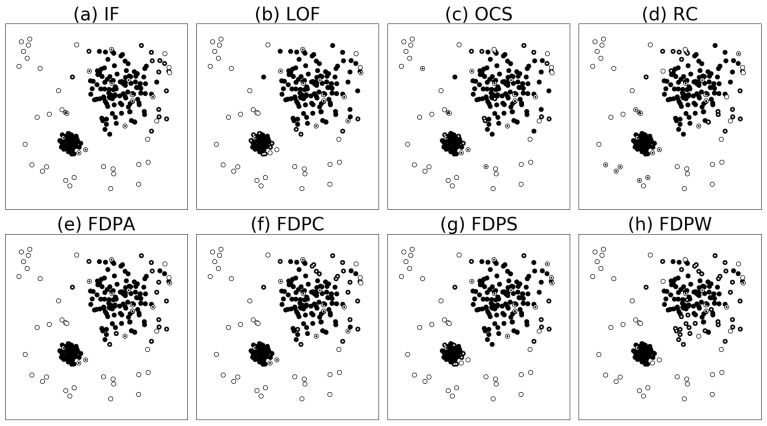
Outlier detection results of the algorithms on synthetic dataset D1. Outliers correct recognized are white points, insiders correct recognized are black points, outliers error recognized are white points with black core, and insiders error recognized are black points with white core.

**Figure 6 sensors-19-01522-f006:**
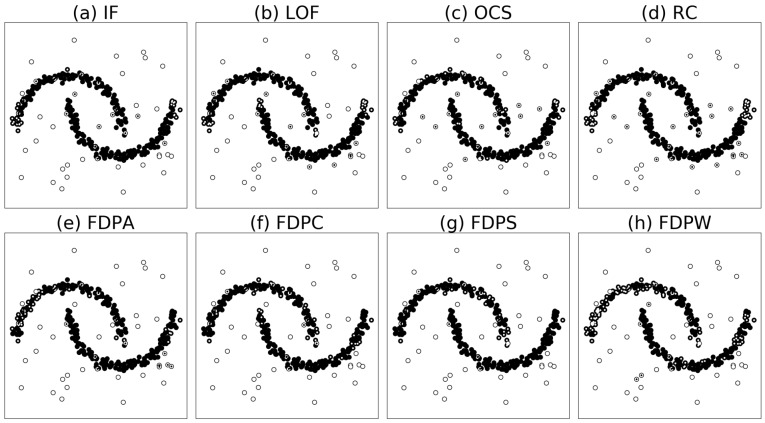
Outlier detection results of the algorithms on synthetic dataset D2. Outliers correct recognized are white points, insiders correct recognized are black points, outliers error recognized are white points with black core, and insiders error recognized are black points with white core.

**Figure 7 sensors-19-01522-f007:**
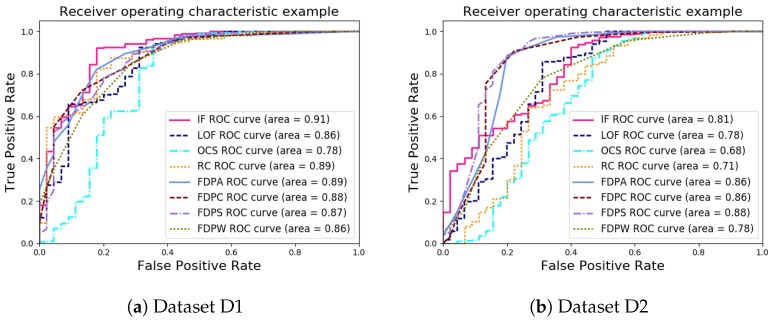
The receiver operating characteristic (ROC) curves and area under curve (AUC) values of synthetic datasets.

**Figure 8 sensors-19-01522-f008:**
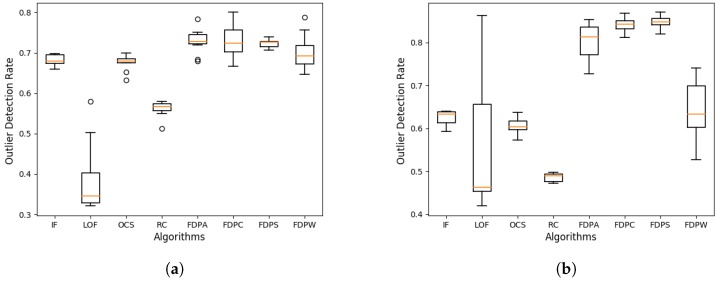
Box plots of the outlier detection rates for dataset series (**a**) D1 and (**b**) D2 with different sized datasets.

**Figure 9 sensors-19-01522-f009:**
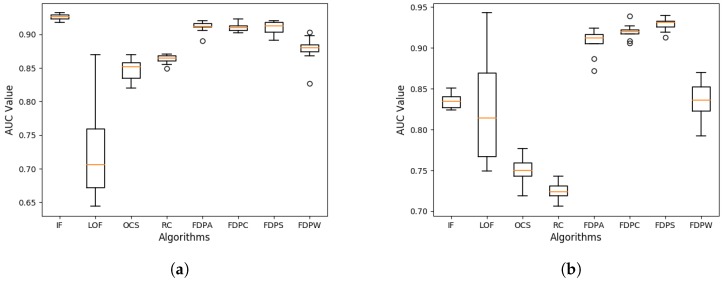
Box plots of the AUCs for dataset series (**a**) D1 and (**b**) D2.

**Figure 10 sensors-19-01522-f010:**
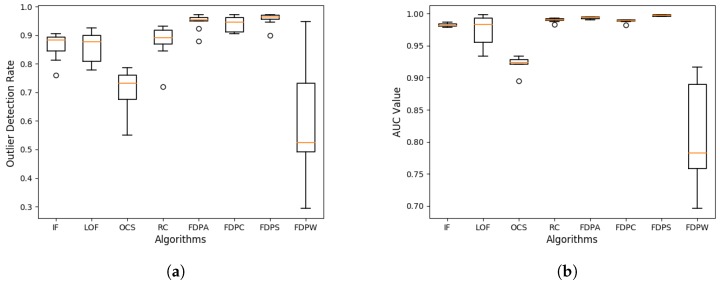
Box plots of the (**a**) outlier detection rates and (**b**) AUCs in the real-world datasets.

**Figure 11 sensors-19-01522-f011:**
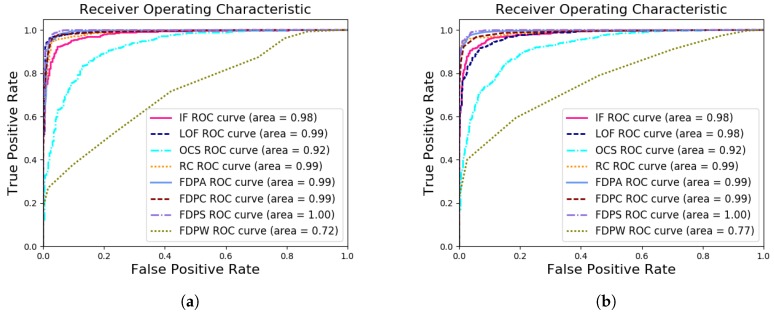
The ROC curves and AUC values for the real-world dataset with (**a**) 300 and (**b**) 600 outliers.

**Table 1 sensors-19-01522-t001:** Summary of the fault detection based on participation degree (FDP) algorithm complexity, where *n* is the number of data items in the dataset, *m* is the maximum number of points in the leaf node, and *m* is much smaller than *n* in practice.

Procedure	Space Complexity	Time Complexity
CreDendTree()	$O(n+m^{2})$	$O(nm+\left(n+\frac{n}{m}\right)\log\frac{n}{m})$
CalParDegree()	O(n)	O(n)
CalThrDegree()	O(n)	O(n)
DetectOutlier()	O(n)	O(n)
**Summary**	O(n)	$O(n\log^{2}n)$

**Table 2 sensors-19-01522-t002:** Summary of synthetic datasets used in experiments.

Dataset	Cluster	Items	Outliers (%)
D1	Two blobs with different radii	300	45 (15%)
D2	Two half circles with the same radius	300	45 (15%)

**Table 3 sensors-19-01522-t003:** The outlier detection rate of isolation forest (IF), local outlier factor (LOF), one-class SVM (OCS), robust covariance (RC), FDP average (FDPA), FDP complete (FDPC), FDP single (FDPS), and FDP Ward’s (FDPW) algorithms in the synthetic dataset experiments. The highest value of the FDP algorithms and the highest value of the comparison algorithms are respectively marked in bold font.

Dataset	IF	LOF	OCS	RC	FDPA	FDPC	FDPS	FDPW
D1	**0.667**	0.644	0.622	0.622	0.644	**0.711**	0.622	**0.711**
D2	**0.600**	0.533	0.511	0.467	**0.800**	0.778	0.756	0.689

**Table 4 sensors-19-01522-t004:** Average time cost (in seconds) of dataset series, data sizes ranged from 1000 to 10,000. The best time cost of the FDP algorithms and the best time cost of the comparison algorithms are respectively marked in bold font.

Series	IF	LOF	OCS	RC	FDPA	FDPC	FDPS	FDPW
D1	0.382	**0.050**	0.279	1.772	1.301	1.200	**0.516**	1.263
D2	0.416	**0.056**	0.364	1.748	1.691	1.530	**0.714**	1.743

**Table 5 sensors-19-01522-t005:** The outlier detection rates of real-world experiments. The highest value of the FDP algorithms and the highest value of the comparison algorithms are respectively marked in bold font.

Outliers	IF	LOF	OCS	RC	FDPA	FDPC	FDPS	FDPW
100	0.760	**0.900**	0.550	0.720	0.880	**0.910**	0.900	0.510
200	0.835	**0.925**	0.635	0.845	0.950	0.915	**0.955**	0.485
300	0.813	**0.907**	0.667	0.863	0.923	**0.950**	0.947	0.293
400	0.878	**0.900**	0.703	0.885	0.958	0.905	**0.963**	0.788
500	0.892	**0.894**	0.726	0.888	**0.968**	0.966	0.966	0.510
600	0.885	0.862	0.740	**0.895**	0.952	0.943	**0.960**	0.540
700	0.893	0.834	0.749	**0.914**	**0.971**	0.906	0.970	0.419
800	0.905	0.801	0.765	**0.919**	0.963	**0.971**	**0.971**	0.743
900	0.883	0.796	0.772	**0.924**	0.963	0.949	**0.971**	0.948
1000	0.906	0.778	0.787	**0.931**	0.950	0.966	**0.967**	0.703
Average	0.865	0.860	0.709	**0.878**	0.948	0.938	**0.957**	0.594

**Table 6 sensors-19-01522-t006:** The AUCs of the real-world datasets used in the experiments. The highest value of the FDP algorithms and the highest value of the comparison algorithms are respectively marked in bold font.

Outliers	IF	LOF	OCS	RC	FDPA	FDPC	FDPS	FDPW
100	0.981	**0.998**	0.895	0.988	0.993	0.990	**0.996**	0.917
200	0.980	**0.995**	0.931	0.983	0.994	0.990	**0.996**	0.856
300	0.981	**0.993**	0.925	0.989	0.992	0.990	**0.996**	0.719
400	0.987	**0.993**	0.929	0.993	0.995	0.989	**0.998**	0.901
500	0.986	0.989	0.934	**0.990**	0.995	0.989	**0.998**	0.794
600	0.982	0.978	0.923	**0.992**	0.995	0.990	**0.998**	0.770
700	0.982	0.968	0.921	**0.992**	0.994	0.982	**0.997**	0.696
800	0.985	0.951	0.925	**0.992**	0.992	0.990	**0.998**	0.771
900	0.979	0.949	0.921	**0.991**	0.994	0.988	**0.998**	0.903
1000	0.980	0.934	0.923	**0.989**	0.990	0.988	**0.996**	0.755
Average	0.982	0.975	0.923	**0.990**	0.993	0.989	**0.997**	0.808

**Table 7 sensors-19-01522-t007:** The time cost (in seconds) of real-world datasets used in the experiments.

Outliers	IF	LOF	OCS	RC	FDPA	FDPC	FDPS	FDPW
100	0.377	0.061	0.030	1.167	0.123	0.088	0.071	0.110
200	0.360	0.059	0.022	1.278	0.125	0.101	0.106	0.115
300	0.340	0.079	0.032	1.328	0.134	0.095	0.095	0.126
400	0.341	0.071	0.047	1.263	0.157	0.113	0.055	0.218
500	0.350	0.086	0.063	1.475	0.160	0.122	0.056	0.220
600	0.340	0.083	0.080	1.084	0.176	0.128	0.059	0.254
700	0.361	0.087	0.099	1.081	0.166	0.140	0.062	0.237
800	0.353	0.100	0.112	1.066	0.204	0.143	0.063	0.265
900	0.379	0.098	0.130	1.065	0.194	0.154	0.068	0.275
1000	0.355	0.102	0.149	1.024	0.233	0.162	0.077	0.307
Average	0.355	0.083	0.076	1.183	0.167	0.125	0.071	0.213

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
