# Peer review of "A Participation Degree-Based Fault Detection Method for Wireless Sensor Networks"

_sensors, 2019, doi:10.3390/s19071522_

Reviewer 1 Report

Regarding the paper entitled "FDP: A Novel Fault Detection Method Based on Participation Degree for WSNs ". The paper is very difficult to read. I recommend reviewing the English with a professional in academic writing.

*It is necessary to modify the title. Is quite wear to see two acronyms, which makes the information difficult to understand.

*The abstract is too long and have not necessary information that can be included in the introduction. For instance, a good beginning could from the sentence " In this paper...".

*I Can't understand this sentence

"which exploits the  relationships between instances to an extent that is not feasible in other techniques"

* Data-based anomaly detection methods can be also achieved by principal component analysis,  neuronal networks, among others, for instance, consider:

https://doi.org/10.1016/j.ifacol.2018.09.604

* The introduction is too long and considers information that is not necessary for the paper, for example, instead of classification, the authors must consider to write the paper as a history with well-defined ideas.

*The rest of the paper is not well written also. I strongly recommend to the authors a careful revision of the paper and also reorganize the paper by including only the necessary information.

*Before include different algorithms, the author's cand include a schematic in order to show the connection among them.

*Improve the conclusions

Author Response

Dear Editors and Reviewers,

We would like to thank you for your time and valuable feedback.

Your comments have helped us to greatly improve the manuscript. We have made significant changes to the paper and other smaller edits. To highlight the changes that we have made in the manuscript, we use the red color for all new text. Below, we provide detailed response to the reviewers’ comments, in which the citation of references is consistent with those in the manuscript. We appreciate each reviewer’s comments and effort in terms of helping us improve the paper. We have done our best to address each comment and respond accordingly and would be glad to submit any additional information if needed.

Thank you again for all your effort on behalf of our submission.

Sincerely, Wei Zhang

Please check the response in the attachment.

Reviewer 2 Report

This manuscript proposed a novel method based on participation degree (FDP) for the fault detection in wireless sensor networks. Compared with existing techniques, the proposed method does not need the data training process and can detect the global outliers without local cluster influence. Finally, the experimental data verified the capacity of the proposed method via the comparison with four commonly used methods. Overall, the topic of this study is novel and interesting. The manuscript is well organised and written. I suggest that it can be considered to be published in Sensors if the authors can well address the following comments.

1. In introduction, please try to reduce the length of introduction of existing anomaly detection techniques, and just give a brief summary.

2. In algorithm complexity analysis, I suggest that a table can be used to summarise the different procedures of the FDP algorithm.

3. More future research should be included in Conclusions.

4. There are several typos that affect the quality of the manuscript. Please revise them.

Author Response

Dear Editors and Reviewers,

We would like to thank you for your time and valuable feedback.

Your comments have helped us to greatly improve the manuscript. We have made significant changes to the paper and other smaller edits. To highlight the changes that we have made in the manuscript, we use the red color for all new text. Below, we provide detailed response to the reviewers’ comments, in which the citation of references is consistent with those in the manuscript. We appreciate each reviewer’s comments and effort in terms of helping us improve the paper. We have done our best to address each comment and respond accordingly and would be glad to submit any additional information if needed.

Thank you again for all your effort on behalf of our submission.

Sincerely, Wei Zhang

Please check the response in the attachment.

Round  2

Reviewer 1 Report

I think that the paper can be accepted in the present form. I have just a comment: Use NOVEL in a title is quite dangerous. I don't think that the approach 

is novel. In fact, there are many fault detection methods bases od data-driven techniques.  The authors must modify the title.

Author Response

Dear Editors and Reviewers,

We would like to thank you for your time and valuable feedback.

Your comments have helped us to greatly improve the manuscript. We have changed the title of the paper in this minor revision and highlighted it in red color.

We appreciate each reviewer’s comments and effort in terms of helping us improve the paper. Thank you again for all your effort on behalf of our submission.

Sincerely, Wei Zhang

————————————————————-

Reviewer 1

1. I think that the paper can be accepted in the present form. I have just a comment:

Use NOVEL in a title is quite dangerous. I don’t think that the approach is novel. In fact, there are many fault detection methods based on data-driven techniques. The authors must modify the title.

Response: Thank you very much for your insightful comments. We are sorry for this in our previous manuscript. We have deleted ”novel” from the title and have modified the title to “A Participation Degree Based Fault Detection Method for Wireless Sensor Networks

Thank you again for your time and efforts to help improve the manuscript. —————————————————————-
